# Nutritional Composition, In Vitro Starch Digestibility and Antioxidant Activities of Composite Flour Made from Wheat and Mature, Unripe Pawpaw (*Carica papaya*) Fruit Flour

**DOI:** 10.3390/nu14224821

**Published:** 2022-11-14

**Authors:** Adeyemi A. Adeyanju, Oluwaseun P. Bamidele

**Affiliations:** 1Department of Food Science and Microbiology, Landmark University, Omu-Aran 1001, Nigeria; 2Department of Food Science and Technology, University of Venda, Thohoyandou 0950, South Africa

**Keywords:** resistance starch, fat acidity, pasting, healthy foods, storage

## Abstract

Due to the rise in the number of people suffering from diet-related noncommunicable diseases, major scientific studies have recently been focused on the development of functional foods that are rich sources of resistant starch and bioactive compounds with health-promoting properties. The nutritional composition, in vitro starch digestibility, and antioxidant properties of composite flour derived from wheat and mature, unripe pawpaw fruit flour are all discussed in this study. The proximate composition, functional and pasting properties, in vitro starch digestibility, antioxidant activities and storage stability of the composite flours were determined. When compared to 100% wheat flour, the crude fiber, ash, water absorption capacity, swelling capacity, and bulk density of the composite flours increased by 40.5–63.3%, 209.7–318%, 2–109%, 3–66%, and 28–162%, respectively. Increased addition of mature, unripe pawpaw fruit flour to wheat flour resulted in a rise in the composite flour’s TPC, ABTS, and ORAC values. Comparing the composite flour made with 50% mature, unripe pawpaw fruit flour to 100% wheat flour, the resistant starch and slowly digested starch rose by 2836% and 1321%, respectively. Additionally, compared to 100% wheat flour, the composite flours also demonstrated decreased fat acidity. It can be argued that the composite flour is a good source of resistant starch and bioactive ingredients that can be used in a variety of functional food products.

## 1. Introduction

Diet-related noncommunicable diseases like type II diabetes, cancer, and cardiovascular disease are a major problem worldwide. Estimates indicate that 23.3 million additional people could die from cardiovascular disease by 2030, up from the 17.5 million who did so in 2012 [1]. One of the main types of cardiovascular disease is atherosclerosis. It is believed that the cause of it is oxidative stress, which is brought on by the accumulation of highly reactive oxygen species, such as oxygen free radicals [2]. The current theory holds that oxidative stress triggers a chain of events that cause the body to accumulate cholesterol, which subsequently restricts the arteries leading to the heart, impairing blood flow [3]. Additionally, reactive oxygen species are thought to play a role in the toxicity of food, skin aging, and diseases including cancer and diabetes [4]. It is well known that bioactive substances with antioxidant characteristics could neutralize free radicals in vitro [5]. These bioactive substances are abundant in plant-based foods such as fruits and vegetables. For instance, pawpaw, especially the unripe pawpaw, has been observed to contain a variety of bioactive compounds [6,7]. The health benefits of food may be enhanced by using mature, unripe pawpaw fruit flour as a composite.

Consuming the entire pawpaw fruit (*Carica papaya*) is a great way to get more dietary fibre and reduce constipation in people. Papaya’s fibre content can aid in decreasing high blood cholesterol levels [8]. Papaya’s nutrients can also aid in stopping the oxidation of cholesterol. Along with being nutrient-rich, papaya also has a variety of bioactive phytochemicals with unique structural and functional characteristics that have not yet been completely explored for their potential health advantages [8]. The interactions between the minerals and phytochemicals in papaya’s full fruit, rather than a single “active” ingredient, are what provide the fruit its overall nutritional and health benefits. Papaya can be used as a preventive treatment for atherosclerosis, strokes, heart attacks, and diabetes and can stop cholesterol from oxidizing due to its high antioxidant content [9]. Interestingly, ripe papaya contains more sugar and less fibre than unripe papaya [10]. In the food sector, papaya is available as processed foods such as dried papaya, candies, and ice cream as well as fresh fruits [11]. When mature, unripe fruits like bananas and cashews were employed to make a variety of food products; improved health benefits have been reported [12]. According to reports, adding mature, unripe banana fruit flour to wheat flour while preparing cookies can lower the composite flour’s glycaemic index and increase the food’s nutritional value [13].

One of the most significant staple food crops for many nations is wheat (*Triticum aestivum*), which is imported into Africa [14]. Protein levels in wheat grains range from less than 8% to more than 20%, making them nutrient-rich [15]. The wheat flour is made from wheat (*Triticum aestivum L.*), which is also used to make bread, scones, other confections, noodles, and pasta [16]. In comparison to other cereals, wheat is a preferable option of cereal crop for baked goods due to the presence of gliadin and glutenin [17]. Wheat can be used as the ideal delivery system for a variety of bioactive compounds, enhancing the health benefits of wheat-based food items since it is used to make a wide variety of food products. Though it is not a tropical crop, just a few African nations have the right climatic conditions to support the growth of wheat.

Finding crops that can replace wheat to lessen reliance on imports is vital due to the economic challenge posed by the cost of importation. There has not yet been a perfect match for wheat grain, but research is constantly being done to discover one. Compositing mature, unripe pawpaw fruit flour with wheat flour will reduce the amount of wheat required in confectioneries production and improve the health benefits of such foods. Hence, the study is aimed at making a composite flour from wheat and mature, unripe pawpaw fruit to determine the nutritional composition, in vitro starch digestibility and antioxidant properties of the composite flour.

## 2. Materials and Methods

### 2.1. Raw Materials Collection

The unripe pawpaw fruit (Kamiya variety) was obtained from Teaching and Agricultural Farm OAU (Obafemi Awolowo University), Ile-Ife, Nigeria, while the wheat flour was purchased at the central market in Obafemi Awolowo University.

#### 2.1.1. Processing of Mature, Unripe Pawpaw Fruit to Flour

The mature, unripe pawpaw fruit was washed and peeled manually using a knife. The peeled mature, unripe pawpaw fruit was deseeded, cut into smaller pieces and dried at 50 °C for 72 h using a forced air oven. The dried pieces were milled into flour using a hammer mill (Lab Mill 3100, PerkinElmer, Buckinghamshire, UK) and sieved with a 500-micron sieve. The flour was packaged in ziplock bags and stored at 4 °C for further analysis.

#### 2.1.2. Preparation of the Composite Flour (Mature, Unripe Pawpaw Fruit Flour and Wheat Flour)

The wheat flour and mature, unripe pawpaw fruit flour were mixed at different ratios. The 100% wheat flour serves as the control sample (A). The 90% wheat flour to 10% mature, unripe pawpaw fruit flour serves as sample B, while 80% wheat flour to 20% mature, unripe pawpaw fruit flour is sample C. Samples D, E, and F, respectively, were made with a ratio of 70% wheat flour to 30% mature, unripe pawpaw fruit flour, 60% wheat flour to 40% mature, unripe pawpaw fruit flour, and 50% wheat flour to 50% mature, unripe pawpaw fruit flour. According to our preliminary study, using mature, unripe pawpaw fruit flour more than 50% resulted in cookies that the panelists found to be quite unappealing and were scored very low in terms of overall acceptability; therefore, we chose to stop at 50%. This may be due to a reduction in glutenin and gliadin protein and a higher amount of starch and fibre. The flours were homogenized properly and packed in ziplock bags and stored in a refrigerator (4 °C) for further analyses.

### 2.2. Proximate Composition of the Composite Flour (Wheat and Mature, Unripe Pawpaw Fruit Flour)

Using the AOAC method [18], the proximate composition of the composite flour and the control sample was determined (methods 925, 10, 65.17, 974, 24, 992.16). The carbohydrate content was estimated by deducting the moisture (%), protein (%), ash (%), and fat (%), and fibre (%) from 100%.

### 2.3. Functional Properties of the Composite Flour (Wheat and Mature, Unripe Pawpaw Fruit Flour)

The Bamidele et al.’s [19] modified method was used to test the functional parameters of the composite flour samples (water absorption capacity (WAC), swelling capacity (SC), and bulk density (BD)). About 30 mL of hot water at 70 °C were added to 2 g of the sample (*W*1) inside the centrifuge tube. The sample was vortexed for 10 min, followed by another 10 min of rest. The suspension was centrifuged at 4100× *g* for 15 min (25 °C). The residue-filled tube was weighed after the supernatant was decanted (*W*2). The formula below was used to calculate the WAC.
(1)$$\mathrm{WAC}=\frac{W2-W1}{Weight\;of\;the\;sample}$$

The swelling capacity of the composite flour sample was evaluated using a modified method of Bamidele et al. [19]. Exactly 2 g of the composite flour were weighed in the centrifuge tube, 30 mL of water were added, and the mixture was then heated to 70 °C for 15 min. Centrifugation of the cooled slurry took place at 4100× *g* for 15 min at 25 °C. The centrifuge tube’s only remaining portion was dried in a hot air oven (50 °C) for 30 min before being weighed after the supernatant was removed. The bulk density was calculated by weighing precisely 10 g of the composite flour sample into a 25 mL graduated cylinder and gently tapping the cylinder on the lab bench until no change in volume was noticed. The sample’s final volume was measured, and the difference between the two was computed to obtain the bulk density, which was then expressed in g/mL. The sample’s water solubility index was calculated using the proportion of dry particles that was recovered from the supernatant after drying overnight in an oven at 100 °C.

### 2.4. Extraction of Free Phenolics

For the free phenolics extraction, 1 g of each flour sample was extracted with 10 mL of MeOH/HCl (85:15) by magnetic stirring for 2 h. The extract was then centrifuged using a Rotanta 460R Hettiech (Hettiech, Cape Town, South Africa) at 2000× *g* for 10 min, the supernatant was separated, and the residue was rinsed four times each with 10 mL MeOH/HCl and centrifuged as above. The supernatants were pooled together and stored at 5 °C in the dark until analysed.

### 2.5. Determination of Total Phenolic Content (TPC) of the Samples

A 100 µL aliquot of each extract from composite flour (wheat flour and mature, unripe pawpaw fruit flour), catechin standard or blank was separately added to a 2 mL Eppendorf microtube in duplicate. To each microtube, 200 µL of 10% Folin–Ciocalteu (F–C) reagent was added, and the mixture was thoroughly vortexed. Subsequently, 800 µL of 700 mM Na_2_CO_3_ were added to each tube. The reactants were incubated in the dark at room temperature for 2 h. Afterwards, 200 µL of the reaction mixture were transferred from the assay tube to a clear 96-well microplate and absorbance was read at 750 nm. The catechin standard curve was used to calculate the total phenolic content, which was expressed as milligram catechin equivalent per gram (mg CE/g) of flour on a dry basis.

### 2.6. Determination of Radical Scavenging Activity (ABTS^+^) of the Samples

The ABTS^+^ radical scavenging capacity of the aliquot of each extract from the composite flour (wheat flour and mature, unripe pawpaw fruit flour) was measured using the method of Apea-Bah et al. [20]. The working solution of 0.01 M (ABTS+) was prepared by mixing 2 mL of ABST ^+^ stock solution of 0.01 M with 58 mL of PBS (pH 6.9). The mixture was left in the dark for 12–16 h. The mixture of the aliquot extracted from the composite flour and the working solution was left in the dark for another 30 min before the absorbance was read at 750 nm. Trolox solution served as the standard and results were reported as μmol TE/g db.

### 2.7. Oxygen Radical Absorbance Capacity (ORAC) Determination of the Flour Samples

The ORAC of the composite flour sample was measured based on the modified method of Ou et al. [21]. The fluorescein stock solution (0.88 mM) was prepared in PBS (pH 6.9). The working solution was prepared by diluting fluorescein stock in 100,000×, yielding a fluorescein concentration of 8.82 nm. About 165 μL of working solution (fluorescein) were added to a 10 μL sample followed by 25 μL of AAPH (240.74 mM), yielding final concentrations of 7.26 nM and 30.13 mM for fluorescein and AAPH, respectively. A 10 μL volume of each extract was added to the wells of a 96-well microplate followed by 165 μL of fluorescein and 25 μL of AAPH. The solutions were mixed, and fluorescence measured at 37 °C, every 2 min for 2 h at an excitation wavelength of 485 nm and an emission wavelength of 520 nm employing a fluorescence plate reader (FLUOstar Omega, BMG LABTECH, Offenburg, Germany). The result was calculated using the net area under the decay curve (AUC) and expressed as μmol TE/g.

### 2.8. Pasting Properties of Composite Flour (Wheat and Mature, Unripe Pawpaw Fruit Flour)

A Rapid Visco Analyser (RVA) series 4 (New Port Scientific, Warriewood, NSW, Australia) was used for the pasting profile of the composite flour. The sample was placed into the canister weighing around 3 g, and twenty-five (25) mL of distilled water was then added. A paddle was placed inside the canister (centrally onto the paddle coupling), then inserted into the RVA machine. The measurement cycle was initiated by pressing the motor tower of the instrument and a 14 min profile was applied. The time–temperature regime used was idle at 50 °C for 1 min, heated from 50 °C to 91 °C in 4 min, then held at 91 °C for 3 min. The sample was subsequently cooled to 50 °C for a 4 min period, followed by a period of 2 min where the temperature was controlled at 50 °C [21]. All measurements were taken in triplicate.

### 2.9. In Vitro Starch Digestibility of the Composite Flour Made from Wheat and Mature, Unripe Pawpaw Fruit Flour

The in vitro starch digestibility of the composite flour was determined by a modification of the procedure of Englyst et al. [22]. Enzyme solution I was created by dilution of amyl glucosidase solution (0.14 mL) with water to 6.0 mL. A portion (54.0 mL) of the supernatant was then transferred into a flask after centrifuging 12× *g* of porcine pancreatin in 80.0 mL of water for 10 min at 1500 g to create enzyme solution II. Just prior to use, water (4.0 mL), enzyme solution I (6.0 mL), and enzyme solution II (6.0 mL) were combined to create enzyme III (54.0 mL). A composite flour sample (200 mg, db) was put in a polypropylene centrifuge tube (30.0 mL), together with enzyme solution III (5.0 mL) and sodium acetate buffer (pH 5.2). The tube was shaken in a water bath at 37 °C (90 strokes per minute). Following 20 min, 0.5 mL of the digest was added to 20.0 mL of 66% ethanol, which was then combined and centrifuged. The amount of glucose in the supernatant was measured using glucose oxidase, and RDS was calculated by multiplying the amount of glucose (in mg) released by 0.9 and dividing it by 200 mg. The amount of glucose released after 120 min of digestion is known as resistant starch, whereas the amount released between 20 and 120 min is known as slowly digested starch (SDS), and the amount released after 20 min is known as quickly digestible starch (RDS). The following formulas were used to determine how much glucose was produced in 120 min.
(2)$$\mathrm{RDS}\;(\%)=\frac{\left(G20-FG\right)}{TS}\times 100$$
(3)$$\mathrm{SDS}\;(\%)=\frac{\left(G120-FG\right)}{TS}\times 100$$
(4)$$\mathrm{RS}\;(\%)=\frac{TS-\left(RSD-SDS\right)}{TS}\times 100$$
where *FG* = glucose content after 0 min of digestion, *G*20 = glucose content after 20 min of digestion, *G*120 = glucose after 120 min of digestion, *TS* = total content starch.

### 2.10. Flour Fat Acidity

Lipids were extracted from flour samples using petroleum ether (boiling point, 40–60 °C). Fat acidity was analysed by acid-base titration according to the AACC Method 02-02 A [23]. All parameters were determined at least at 0-, 3-, 6- and 9-weeks of flour storage at room temperature (29 °C).

### 2.11. Statistical Analysis

All data obtained were subjected to analysis of variance (ANOVA) using the Statistical Package for Social Science (SPSS; version 21). Significant means were separated using Duncan’s multiple comparison tests at 5% level of probability.

## 3. Results and Discussions

### 3.1. Proximate Composition of the Samples

The proximate composition of 100% wheat flour and the composite flours made from wheat and mature, unripe pawpaw fruit is shown in Table 1. There was no significant difference (*p* > 0.05) in moisture content of all the samples. The proportion of the mature, unripe pawpaw fruit flour (MUPFF), which ranges from 10 to 50%, caused the crude protein content of the composite flours to drop by roughly 16 to 24%. The highest value (12.68%) was in the control sample, which contained only wheat flour, and was followed by composite flour containing 10% MUPF. For the composite flour with 50% MUPF, the lowest crude protein value (9.12%) was found. The pattern for crude fat was the same as for crude protein. The control sample had the highest crude fat value (3.11%), followed by the sample with 10% MUPF (3.01). The least amount of crude fat (1.62%) was found in the composite flour that contained 50% MUPFF. As the proportion of MUPFF added to wheat flour rose from 10% to 50%, the crude fiber and ash content of the composite flours significantly increased (*p* < 0.05). The composite flour containing 50% MUPF had the highest crude fiber value (8.67%), whereas the control sample had the lowest (5.31%). The control sample also had the least amount of ash (0.72%), which was followed by the composite flour containing 10% MUPFF (2.23%). The highest value, 3.01%, was found in the composite flour containing 50% MUPFF. The carbohydrate content of the control sample and the composite flour with 20, 30, 40, and 50% MUPFF did not differ significantly (*p* < 0.05). The least amount of carbohydrate (67.94%) was found in the composite flour containing 10% MUPFF.

The addition of MUPFF to wheat flour brought about variations in the flour samples’ proximate composition. For instance, lower levels of crude protein and crude fat in MUPFF compared to wheat flour led to a reduction in the levels of crude protein and fat in the composite flours compared to 100% wheat flour, while higher levels of crude fiber and ash in the composite flours compared to 100% wheat flour can be attributed to MUPFF’s high levels of crude fiber and ash, as previously reported [24]. This shows that the composite flours may be high in dietary fiber and, as a result, provide a food product with a very low glycaemic index. Low-glycaemic-index flour samples have been associated with high levels of crude fiber [25]. This result is consistent with the earlier work [26] that found that adding 5–20% unripe papaya flour to pancake significantly reduced the amount of glucose released. Additionally, the composite flours’ high ash content may make them a rich source of minerals including calcium, potassium, iron, and zinc that may aid in the prevention of disease and the maintenance of good health [27]. This is also in line with the findings of Okon et al. [24], who asserted that unripe papaya fruit are high in ash and crude fiber.

### 3.2. Functional Properties of the Composite Flour

The functional properties of the composite flour made from wheat flour and unripe pawpaw fruit flour, as determined by water absorption capacity, swelling capacity, bulk density, and water solubility index, are displayed in Table 2. Food materials’ processing behaviour is highly influenced by their functional qualities. Additionally, they aid in deciding what kind of preservation and packaging materials are appropriate for delivering such processed food [28]. For instance, the water absorption capacity, a measurement of the flour samples’ capacity to hold water, ranged across all flours from 117.51% to 244.01%, with 100% wheat flour having the lowest value and the composite flour made up of 50% wheat flour (WF) and 50% mature, unripe pawpaw fruit flour (MUPFF) having the highest value. It is an indication of the amount of water absorbed by the flour to reach the necessary consistency and produce a high-quality finished product. The results further indicated that the water absorption capacity of the flour samples increased significantly (*p* < 0.05) by approximately 33–109% as the levels of the substitution of the WF with MUPFF in the composite flour increased from 10 to 50%, indicating that the inclusion of MUPFF was able to increase the ability of the flour to absorb water. This may be attributed to the presence of more hydrophilic constituents such as polysaccharides or higher starch damage in the MUPFF compared to the 100% WF. As indicated in Table 1, the MUPFF contains more fibre than the WF, which is another possible explanation. According to an earlier study [29], dietary fibre can improve water absorption and is linked to the dilution of gluten, which leads to an increase in the hardness of the texture. The high-water absorption capacity of the composite flour is an indication that the flour will be very useful in products where good viscosity is required.

Swelling capacity has been identified as one of the criteria for a high-quality product. Important characteristics that eventually influence sample consistency include swelling and water absorption capabilities, which depend on the compositional structure of the sample. When preparing food, flours with high water absorption and swelling capacities hold a lot of water. The swelling capacity of the flour samples followed a similar pattern with water absorption capacity, as the value ranged from 127.74 to 173.22 for 100% WF and 50 WF + 50 MUPFF, respectively. The value increased by 3–36% as the proportion of the MUPFF in the composite flour increased from 10 to 50%, indicating that the incorporation of the MUPFF significantly affects the swelling capacity of the flour mixture.

The relatively high fibre content and chemical makeup of the MUPFF, which have the capability to absorb water molecules, may be responsible for the composite flour’s increased swelling capacity when compared to 100% WF [26]. Food qualities including body, thickening, and increased viscosity are impacted by swelling because it alters the hydrodynamic properties of the food [30]. This suggests that when compared to 100% WF, the composite flour will generate a thick, viscous product, which is often desirable in some food products.

The incorporation of MUPFF into WF also enhanced the bulk density of the composite flour as the value increased from 1.53 ± 0.5 (100% WF) to 4.01 ± 0.6 (50 WF + 50 MUPFF), indicating that the value increased by approximately 28–162% as the proportion of the MUPFF in the composite flour increased from 10 to 50%. Bulk density is a measure of a product’s porosity that affects packaging design and may be used to choose the right kind of material for wet processing in the food industry, as well as to determine how to handle materials. High bulk density is a good physical property to consider when assessing the mixing quality of a particular matter [31].

The water solubility index, in contrast to other functional parameters found, significantly (*p* < 0.05) fell from 2.93 ± 0.2 (100% WF) to 1.03 ± 0.2 (50 WF + 50 MUPFF) when the percentage of the MUPFF increased from 10 to 50%. The water solubility index indicates the amount of free polysaccharide or polysaccharide liberated from the granule when too much water is added to the flour and thus shows the degree of starch breakdown [31] The findings suggested that there was less starch breakdown in the MUPFF than in the WF, because the composite flours had a lower water solubility index than the 100% WF.

### 3.3. Total Phenolic Content and Antioxidant Activities of the Samples

In Table 3, the findings of the samples’ total phenolic content (TPC) and antioxidant capacities 2,2′-azinobis-(3-ethylbenzothiazoline-6-sulfonic acid) (ABTS) and oxygen radical absorbance capacity (ORAC) are presented. The total phenolic content of the flour samples ranged from 16.09 ± 0.6 to 27.12 ± 0.3, with 100% WF having the lowest value and the composite flour, which contains 50% WF + 50% UPFF, having the highest value. All the composite flours exhibited significantly higher TPC compared to the 100% WF, indicating the presence of higher phenolic constituents in the MUPFF compared to the WF. As the proportion of the UPFF in the composite flour increased from 10 to 50%, the TPC increased by approximately 13 to 69%. Previous studies [6,7,32] have shown that pawpaw contains a wide range of phytochemicals such as epigallocatechin, epicatechin, gallic acid, catechin, *p*-coumaric acid, β-carotene, vitamin C, procyanidin and quercetin.

The ABTS radical scavenging activity of the flour samples showed that the composite flours exhibited higher radical scavenging activities compared to the 100% WF. As the proportion of the MUPFF in the composite flour increased from 10 to 50%, the ABTS radical scavenging activity increased by 154–531%, indicating that the composite flours may have better protective effect against free radicals compared to 100% WF. Similarly, the oxygen radical absorbance capacity of the composite flours was significantly higher (by 164–455%) compared to the 100% WF. The results clearly showed that the TPC, and radical scavenging activities of the composite flours, increased significantly as the proportion of the MUPFF increased from 10 to 50%, indicating that incorporation of MUPFF into WF may enhance its health-promoting properties. These findings highlight the significance of incorporating MUPFF into WF in food preparation. Since the flour sample with the highest TPC also had the highest radical scavenging activity, the data further suggested that the phenolic components in the flour samples are what give it its radical scavenging properties.

### 3.4. Pasting Properties of Composite Flour from Wheat Flour and Mature, Unripe Pawpaw Fruit Flour

As demonstrated in Table 4, the composite flour blends’ pasting properties differed significantly (*p* < 0.05). Peak viscosity, which is the highest viscosity that develops during or shortly after heating the flour samples and a measure of the starch’s capacity to bind water, ranged from 120.46 ± 1.5 to 202.41 ± 1.6, with 100% WF having the lowest value and the composite flour made up of 50% WF and 50% MUPFF having the highest value. Peak viscosity measurement is essential for determining the cooking and baking qualities of flour. The results clearly indicated that peak viscosity increased significantly (*p* < 0.05) by approximately 18 to 68% as the proportion of MUPFF in the composite flour increased from 10 to 50%, suggesting that the composite flours had higher water binding potential compared to 100% WF. Since proteins and fats are known to reduce peak viscosity through their interaction with starch, the higher peak viscosity seen in the composite flours may be explained by the lower crude fat and protein contents in MUPFF compared to WF. Considering that composite flours have a high peak viscosity, the results indicated that they would yield stiff dough products with good textural quality [33]. This is very important when making amala, a stiff dough product made from wheat flour that is typically consumed with soup.

The trough viscosity is the point during either a heating or cooling process where the viscosity is at its lowest. It measures how well the paste will hold up to disintegration during cooling. The value ranged from 76.46 ± 0.5 (100% WF) to 92.21 ± 0.8 (50% WF + 50% MUPFF). The tendency of the composite flours to disintegrate during cooking is indicated by the much higher trough viscosity seen in the composite flours compared to 100% WF. When compared to 100% WF, the breakdown viscosity of the composite flour was significantly higher (by about 44 to 151%), indicating that 100% WF will be able to withstand heat and shear stress better than the composite flours. This is because the breakdown viscosity is a measure of the stability of the starch and an index of how well the paste will withstand heat and shear stress during cooking. Adebowale et al. [34] found that a sample’s ability to withstand heat and shear stress during cooking decreased with increasing breakdown viscosity. The findings therefore showed that adding MUPFF to WF decreased the composite flour’s resistance to heat and shear stress.

Final viscosity, which reflects a flour’s capacity to transform into a sticky paste after cooking and cooling, is frequently used to describe a starch-based flour’s quality. Additionally, it provides a measurement of the paste’s resistance to shear force during stirring. The final viscosity values ranged from 112.51 ±1.6 to 272.43 ± 1.6, with 100% WF having the lowest value and the 50:50 flour mixture having the highest value, demonstrating that the addition of MUPFF to WF improved the composite flour’s capacity to create a thick paste after heating and cooling. The composite flours showed considerably (*p* < 0.05) higher setback viscosity values (by 107–399%) than the 100% WF, suggesting that the end products made from these flour samples will be less prone to retrogradation and staling. The tendency for retrogradation and staling is less common in flour samples with a higher setback viscosity [35]. Peak time, which measures the minimal temperature and cooking time required to cook flour, ranged from 5.12 to 5.68 min, and did not significantly differ between the flour samples. The fact that the samples’ peak temperatures were essentially the same suggests that cooking virtually all the flour samples will need around the same amount of energy and time.

### 3.5. The In Vitro Starch Digestibility of the Flour Samples

The in vitro starch digestibility of the flour samples is shown in Figure 1 (Appendix A). Rapidly digestible starch (RDS) was highest (97.56%) in the control sample, which was made up entirely of wheat flour, and was followed by composite flour made up of 10% mature, unripe pawpaw fruit flour (79.91%). The least rapidly digested starch (29.22%) was found in composite flour with 50% MUPFF. The result showed that there was a 70% reduction in in vitro starch digestibility when the level of MUPFF in the composite flour was increased to 50% compared to 100% wheat flour. As the proportion of MUPFF in the composite flour increased, so did the slowly digestible starch (SDS) and resistant starch (RS) content. The pasted composite flour sample with 50% MUPFF had the highest SDS value (34.67%), while the pasted control sample (100% wheat flour) had the lowest (2.44%). The result clearly demonstrated that the composite flour with 50% MUPFF had a 1321% higher SDS content than flour made entirely from wheat. Similar to this, the sample containing 50% MUPFF had the highest RS value (36.11%), whereas the control sample had the lowest RS (1.23%), suggesting that the RS increased by 2836% in the composite flour containing 50% MUPFF compared to wheat only flour. The results showed that the addition of MUPFF to wheat flour increased the RS, making the composite flour a better raw material for processing healthy foods, particularly for people with diabetes.

The increase in the RS of the composite flour may be attributed to a higher amount of crude fibre in the MUPFF (Table 1) and the formation of different complexes (amylose-lipids complexes and starch–lipid–protein complexes) during pasting. Amylose–lipid complexes, which have been tagged as RS type III or V [36], are formed when the endogenous lipids in food materials interact with leached-out amylose to form a complex. The pasting process, which involves heating and cooling, may also result in starch–lipid–protein interactions, which could enhance RS levels, as demonstrated by the results of this study. High glycaemic index is caused by a rise in RDS value in food substances, which has been linked to several chronic diet-related noncommunicable diseases (such as diabetes, obesity, and cardiovascular diseases). On the other hand, an increase in RS in food is known to lower the glycaemic index, because the α-amylase enzyme has trouble breaking down the starch, which results in a gradual release of glucose into the blood [37]. This outcome is consistent with the findings of Jomark et al. [26], who found that pancakes made from unripe papaya flour had a higher percentage of undigestible starch.

### 3.6. Flour Fat Acidity during Storage

Figure 2 shows the fat acidity of the flour samples during storage. The fat acidity of all the samples increased with an increase in storage time, though at varying rates. The maximum fat acidity was found in 100% wheat flour, and the acidity level rose as the storage duration increased. The fat acidity did, however, drop as the quantity of MUPFF increased in the composite flours. The fat acidity of composite flour containing 10% MUPFF was less than 20 mg KOH/100 g, whereas that of composite flour containing 20% MUPFF was less than 15 mg KOH/100 g. The least amount of fat acidity—less than 9 mg KOH/100 g—was found in the composite flour that contained 50% MUPFF. Because wheat flour has more crude fat than the composite flours, the enzyme lipase can react with this fat to raise the acidity, which is why the acidity of the 100% wheat flour has increased with time. Although the moisture content of the stored samples was not assessed, an increase in fat acidity has been linked to an increase in the moisture content of stored flour [38]. It is important to note that the fat acidity also increased with storage time in the composite flour; however, the increase was considerably less than what was recorded for 100% wheat flour. This may be due to the small amount of fat that is available for the lipase enzyme to work upon. The composite flours’ low level of fat acidity is a sign that they will be shelf-stable for a long time without developing oxidative rancidity.

The composite flours made from wheat and mature, unripe pawpaw fruit flour are relatively stable for nine weeks of storage in addition to having high-resistant starch, antioxidant activity, and good pasting qualities. This makes the composite flours a healthy raw material for the food industry when producing confectionaries and other functional foods. Appendix A shows the schematic illustration of this study.

## 4. Conclusions

This study highlights the significance of compositing wheat flour with mature, unripe pawpaw fruit. While the composite flour’s levels of crude protein and crude fat dropped, ash, resistant starch, and antioxidant activity significantly increased. The composite flour also offers enhanced storage stability and superior pasting properties. It can be concluded that adding mature, unripe pawpaw fruit flour to wheat flour may aid in the development of a variety of functional foods, especially for those living with diabetes.

## Figures and Tables

**Figure 1 nutrients-14-04821-f001:**
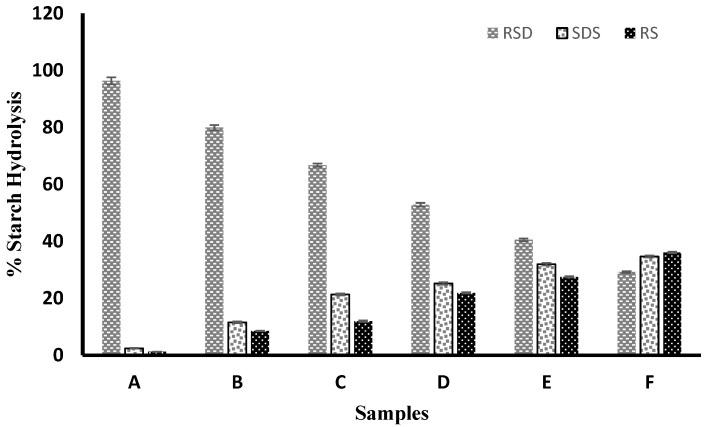
In vitro starch digestibility of composite flour made from wheat and unripe pawpaw fruit flour. Keys: A is 100% wheat flour; B is 90% wheat and 10% mature, unripe pawpaw fruit flour; C is 80% wheat and 20% mature, unripe pawpaw fruit flour; D is 70% wheat and 30% mature, unripe pawpaw fruit flour; E is 60% wheat and 40% mature, unripe pawpaw fruit flour; F is 50% wheat and 50% mature, unripe pawpaw fruit flour.

**Figure 2 nutrients-14-04821-f002:**
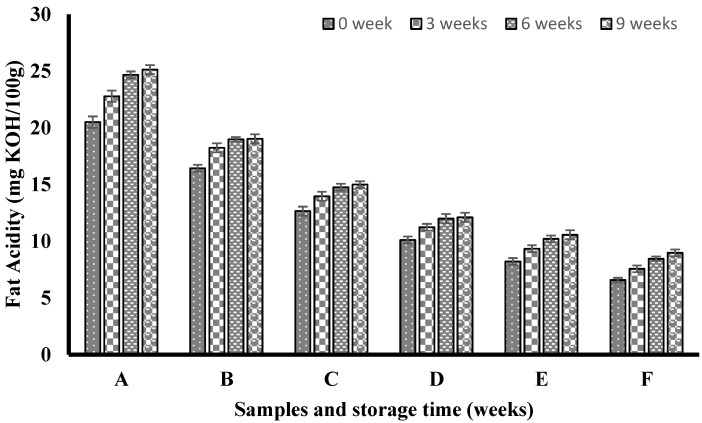
Fat acidity of control and composite flour made from mature, unripe pawpaw fruit flour and wheat flour stored for nine weeks. Keys: A is 100% wheat flour; B is 90% wheat and 10% mature, unripe pawpaw fruit flour; C is 80% wheat and 20% mature, unripe pawpaw fruit flour; D is 70% wheat and 30% mature, unripe pawpaw fruit flour; E is 60% wheat and 40% mature, unripe pawpaw fruit flour; F is 50% wheat and 50% mature, unripe pawpaw fruit flour.

**Table 1 nutrients-14-04821-t001:** Proximate composition of composite flour from wheat flour and unripe pawpaw fruit flour (%) db.

Samples (%)	Moisture	Crude Protein	Crude Fat	Crude Fiber	Ash	Carbohydrate
100% Wheat flour	7.20 ^a^ ± 0.2	12.68 ^c^ ± 0.5	3.11 ^b^ ± 0.1	5.31 ^a^ ± 0.4	0.72 ^a^ ± 0.1	70.80 ^b^ ± 0.9
90 WF + 10 MUPFF	7.38 ^a^ ± 0.3	11.98 ^c^ ± 0.6	3.01 ^b^ ± 0.3	7.46 ^a^ ± 0.4	1.23 ^b^ ± 0.1	67.94 ^a^ ± 0.8
80 WF + 20 MUPFF	7.51 ^a^ ± 0.3	10.01 ^b^ ± 0.6	2.64 ^b^ ± 0.2	7.76 ^b^ ± 0.3	2.03 ^c^ ± 0.1	69.55 ^b^ ± 0.5
70 WF + 30 MUPFF	7.53 ^a^ ± 0.5	9.62 ^b^ ± 0.5	2.02 ^a^ ± 0.3	8.02 ^b^ ± 0.5	2.67 ^c^ ± 0.2	70.14 ^b^ ± 0.7
60 WF + 40 MUPFF	7.54 ^a^ ± 0.4	9.32 ^a^ ± 0.4	1.67 ^a^ ± 0.2	8.35 ^c^ ± 0.3	2.96 ^c^ ± 0.2	70.16 ^b^ ± 0.6
50 WF + 50 MUPFF	7.55 ^a^ ± 0.3	9.12 ^a^ ± 0.5	1.62 ^a^ ± 0.1	8.67 ^c^ ± 0.5	3.01 ^c^ ± 0.3	70.03 ^b^ ± 0.5

Values are means ± standard deviations of replicate determinations (n = 3). Mean values with the same letter in the same column are not significantly (*p* > 0.05) different. WF is wheat flour. MUPFF is mature, unripe pawpaw fruit flour.

**Table 2 nutrients-14-04821-t002:** Functional properties of composite flour from wheat flour and unripe pawpaw fruit flour.

Samples (%)	Water Absorption Capacity (%)	Swelling Capacity (%)	Bulk Density (g/mL)	Water Solubility Index (%)	
100% Wheat flour	117.51 ^a^ ± 1.5	127.74 ^a^ ± 1.8	1.53 ^a^ ± 0.5	2.93 ^f^ ± 0.2	
90 WF + 10 MUPFF	119.21 ^b^ ± 1.6	131.22 ^b^ ± 1.6	1.96 ^b^ ± 0.6	2.53 ^d^ ± 0.3	
80 WF + 20 MUPFF	158.11 ^c^ ± 1.5	139.29 ^c^ ± 1.4	2.10 ^c^ ± 0.8	2.13 ^c^ ± 0.5	
70 WF + 30 MUPFF	193.12 ^d^ ± 1.6	145.23 ^d^ ± 1.4	2.62 ^d^ ± 0.6	1.56 ^c^ ± 0.3	
60 WF + 40 MUPFF	210.12 ^e^ ± 1.7	153.41 ^e^ ± 1.5	3.12 ^e^ ± 0.5	1.23 ^b^ ± 0.3	
50 WF + 50 MUPFF	244.01 ^f^ ± 1.9	173.22 ^f^ ± 1.6	4.01 ^f^ ± 0.6	1.03 ^a^ ± 0.2	

Values are means ± standard deviations of replicate determinations (n = 3). Mean values with the same letter in the same column are not significantly (*p* > 0.05) different. WF is wheat flour. MUPFF is mature, unripe pawpaw fruit flour.

**Table 3 nutrients-14-04821-t003:** Total phenolic content and antioxidant properties of composite flour from wheat flour and unripe pawpaw fruit flour.

Samples (%)	TPC (mg CE/g)	ABTS (µmol TE/g)	ORAC (µmol TE/g)
100% Wheat flour	16.09 ^a^ ± 0.6	1.62 ^a^ ± 0.2	1.12 ^a^ ± 0.1
90 WF + 10 MUPFF	18.26 ^b^ ± 0.5	4.11 ^b^ ± 0.1	2.96 ^b^ ± 0.1
80 WF + 20 MUPFF	20.15 ^c^ ± 0.5	6.32 ^c^ ± 0.1	3.13 ^c^ ± 0.2
70 WF + 30 MUPFF	25.44 ^d^ ± 0.4	7.15 ^d^ ± 0.1	4.65 ^d^ ± 0.3
60 WF + 40 MUPFF	25.69 ^d^ ± 0.4	8.86 ^e^ ± 0.2	5.96 ^e^ ± 0.2
50 WF + 50 MUPFF	27.12 ^e^ ± 0.3	10.22 ^f^ ± 0.2	6.22 ^f^ ± 0.3

Values are means ± standard deviations of replicate determinations (n = 3). Mean values with the same letter in the same column are not significantly (*p* > 0.05) different. WF is wheat flour. MUPFF is mature, unripe pawpaw fruit flour.

**Table 4 nutrients-14-04821-t004:** Pasting properties of composite flour from wheat flour and unripe pawpaw fruit flour.

Samples (%)	Peak Viscosity (RVU)	Trough (RVU)	Breakdown Viscosity (RVU)	Final Viscosity (RVU)	Setback Viscosity (RVU)	Peak Time (Min)	Peak Temperature (°C)
100% Wheat flour	120.46 ^a^ ± 1.5	76.46 ^a^ ± 0.5	44.00 ^a^ ± 0.6	112.51 ^a^ ± 1.6	36.12 ^a^ ± 0.3	5.12 ^a^ ± 0.1	81.26 ^a^ ± 0.6
90 WF + 10 MUPFF	142.02 ^b^ ± 1.6	78.46 ^b^ ± 0.6	63.56 ^b^ ± 0.5	153.21 ^b^ ± 1.5	74.75 ^b^ ± 0.6	5.53 ^a^ ± 0.2	84.12 ^b^ ± 0.5
80 WF + 20 MUPFF	162.62 ^c^ ± 1.6	82.44 ^c^ ± 0.8	80.18 ^c^ ± 1.4	175.21 ^c^ ± 1.7	92.77 ^c^ ± 0.8	5.61 ^a^ ± 0.1	84.22 ^b^ ± 0.7
70 WF + 30 MUPFF	181.22 ^d^ ± 1.5	85.66 ^d^ ± 0.7	96.56 ^d^ ± 1.5	192.31 ^d^ ± 1.6	106.65 ^d^ ± 0.9	5.54 ^a^ ± 0.2	84.25 ^b^ ± 0.6
60 WF + 40 MUPFF	193.41 ^e^ ± 1.5	89.76 ^e^ ± 0.6	103.65 ^e^ ± 2.6	221.41 ^e^ ± 1.5	131.65 ^e^ ± 0.9	5.56 ^a^ ± 0.1	84.56 ^b^ ± 0.5
50 WF + 50 MUPFF	202.41 ^f^ ± 1.6	92.21 ^f^ ± 0.8	110.22 ^f^ ± 1.6	272.43 ^f^ ± 1.6	180.20 ^f^ ± 1.2	5.68 ^a^ ± 0.1	84.67 ^b^ ± 0.5

Values are means ± standard deviations of replicate determinations (n = 3). Mean values with the same letter in the same column are not significantly (*p* > 0.05) different. WF is wheat flour. MUPFF is mature, unripe pawpaw fruit flour.

## Data Availability

The data for this study is available and can be given on request from the corresponding author.

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
