# Peer review of "Nutritional Composition, In Vitro Starch Digestibility and Antioxidant Activities of Composite Flour Made from Wheat and Mature, Unripe Pawpaw (Carica papaya) Fruit Flour"

_nutrients, 2022, doi:10.3390/nu14224821_

Round 1

Reviewer 1 Report

In this manuscript the authors studied the proximate composition, functional and pasting properties, in-vitro starch digestibility, antioxidant activity, and storage stability of the composite flour derived from wheat and mature, unripe pawpaw fruit flour. It is an interesting topic to varieties of readers.

(1) Compositing mature unripe pawpaw fruit with wheat flour to make a composite flour, seems reducing the required wheat amount, but it may be not low cost in mass production.

(2) A schematic illustration of the whole study may be provided for the readers to better understand the content.

(3) Table 1 page 7, why the ash content drastically increases from 0.72 to 2.23, while changes slowly from 90 WF to 50 WF?

(4) What is the fat acidity kind in the composite flour to be shelf stable? Such as the carbon atom numbers, unsaturated bonds.

(4) The parameters are enough to illustrate the good properties of the composite flour. But if some pictures could be provided on some physical properties, such as viscosity, swelling capacity, and water solubility, it will be much better.

Author Response

We will like to thank the reviewer for painstakingly going through the manuscript. Also, all the comments are attended to accordingly and the response is attached. 

Reviewer 2 Report

Due to the rise in the number of people suffering from diet-related non-communicable diseases, major scientific studies have recently been focused on the development of functional foods that are rich sources of resistant starch and bioactive compounds with health promoting properties. Unfortunately, many of our staple foods, such as rice and wheat, increase the taste while removing bioactive ingredients such as dietary fiber and minerals. Upon their consumption, the released sugars are mostly absorbed, categorising these kinds of food such rice and wheat as a high glycemic index food, which increases the risk of people suffering from obesity, diabetes and other chronic diseases. It has been demonstrated that protein interactions with starch, fibre and phenolic compound reduced hydrolysis of carbohydrates. Therefore, understanding the effects of ingredients on the digestibility of food could permit to achieve a controlled glucose release from carbohydrates based foods. In this manuscript, the nutritional composition, in-vitro starch digestibility, and antioxidant properties of composite flour derived from wheat and mature, unripe pawpaw fruit flour are all discussed, which provides an important source of raw materials for low GI foods. This research has positive practical significance. The following comments need to be further supplemented and improved

1.      The concentration added in the test of the manuscript needs to be explained and explained.

From the results of this manuscript, with the increase of the proportion of papaya flour, its starch digestion also decreased, even if it was the highest addition concentration set (50%  mature, unripe pawpaw flour) by the author of this manuscript. It seems that 50% addition does not achieve the maximum or best effect in reducing the starch digestibility and other characteristics of wheat flour in vitro. Therefore, from the effect of reducing the starch digestibility of wheat flour, the study of this manuscript is incomplete. So why does the author choose 50% as the maximum concentration? Whether the added synthetic materials are palatable and more economical, or others,

2.      The scientific value and level of the manuscript need to be improved.

 For example, it is necessary to strengthen the analysis of the advantages and disadvantages of this addition method and other addition methods. It is good taste, low cost and others. It is significant to increase the scientific value and universality of the manuscript.

3.      Manuscript writing needs to be further standardized

The letter “p” of significance analysis is generally expressed in italics, and the full text of the manuscript is in regular script, which requires the author to check and revise the full text.

Author Response

We thank the reviewer for the constructive comments to make the manuscript better. All the comments are attended to accordingly.

Round 2

Reviewer 2 Report

I carefully compared the revisions of the two editions, and I found that the author almost did not modify the original text of the manuscript, but made some simple replies in response to my letter. Personally, I think that the revision is hasty and not too serious. I also think that my comments about the original manuscript are not only what I want to know, but also what more readers may be interested in. Therefore, these opinions need to be reflected in your revised manuscript. Why choose a 50% ratio (That is the composite flour made with 50% mature, unripe pawpaw flour to 100% wheat flour), If the space is limited, the preliminary test results can be listed in the supplementary documents to explain the scientificity and rationality of selecting 50% of the proportion for this study.

Therefore, I do not think that the author's revision of this edition of this manuscript is sufficient, and the revised manuscript need to supplement it based on my comments on the first edition of the manuscript. In addition, in order to better reflect the author's revision, please mark the revised part in different colors, or show the revised parts with the page number and number of lines of the revised manuscript in the letter of response to the comments, so as to facilitate the reviewer's check.

Author Response

Attached is our response to the reviewer's comment
